# Nurturing Deep Tech to Solve Social Problems: Learning from COVID-19 mRNA Vaccine Development

**DOI:** 10.3390/pathogens11121469

**Published:** 2022-12-05

**Authors:** Ryo Okuyama

**Affiliations:** College of International Management, Ritsumeikan Asia Pacific University, Beppu 874-8577, Japan; ryooku@apu.ac.jp

**Keywords:** mRNA vaccine, COVID-19, technology, development, clinical trial, administration, investment

## Abstract

In mRNA vaccines against COVID-19, a new technology that had never been used for approved drugs was applied and succeeded in rapid clinical use. The development and application of new technologies are critical to solving emerging public health problems therefore it is important to understand which factors enabled the rapid development of the COVID-19 mRNA vaccines. This review discusses administrative and technological aspects of rapid vaccine development. In the technological aspects, I carefully examined the technology and clinical development histories of BioNTech and Moderna by searching their publication, patent application and clinical trials. Compared to the case of Japanese company that has not succeeded in the rapid development of mRNA vaccine, years of in-depth technology research and clinical development experience with other diseases and viruses were found to have enhanced BioNTech and Moderna’s technological readiness and contributed to rapid vaccine development against COVID-19 in addition to government administrative support. An aspect of the investments that supported the long-term research and development of mRNA vaccines is also discussed.

## 1. Introduction

The word VUCA stands for volatility, uncertainty, complexity, and ambiguity. Recent years have been called the era of VUCA. The rapid progress of globalization and the information society, as well as various social issues such as global environmental changes and the aging of society, have caused many unexpected events to occur, making it difficult to predict the future. In such an increasingly complex society, countries and companies are required to have strategies and resilience to flexibly respond to changes in political, economic, and social conditions [1,2]. These social issues are solved partly by “deep tech”, which is an advanced science and technology represented by life science, energy, clean technology, computer science, materials science, etc. [3]. Deep tech is mainly researched at universities, then incubated in deep tech start-ups and used in a variety of products and services [4]. The development of products based on these advanced technologies not only contributes to industrial innovation but also plays a role in solving various social problems facing humankind [3,4].

The LNP-mRNA technology is an advanced technology that had never before been applied to a marketed drug [5]. This technology is a protein-coding mRNA that is modified to reduce side effects in humans and encapsulated in a lipid bilayer membrane to increase stability [5]. Compared to conventional vaccines, the lead time for synthesis and manufacturing of mRNA vaccines is shorter, enabling speedy development and clinical application [6]. The development of mRNA vaccines has successfully provided the opportunity to prevent infection in people around the world in a short period of time. The development of a new drug usually takes 10–15 years, but the mRNA vaccines developed by BioNTech and Moderna, respectively, were approved for emergency use by the FDA in December 2020, only one year after the first COVID-19 infected patient was identified [7]. However, not all companies developing mRNA vaccines against COVID-19 have been as successful in the short term as BioNTech and Moderna [8]. In Japan, Daiichi is developing an mRNA vaccine against COVID-19, but they are still stuck in the clinical trials. Therefore, it is worth examining how BioNTech and Moderna were able to develop mRNA vaccines in such a rapid manner. This will have implications for the future development of advanced technologies to confront emergent public health challenges and will provide useful information for readers working on pathogens.

This review examines the factors contributing to the rapid development of BioNTech and Moderna’s COVID-19 mRNA vaccines from three aspects. The first is the administrative aspect. New drug development is heavily influenced by government regulations. Clinical development of new drug candidates must strictly adhere to the pharmaceutical regulations set by the regulatory authorities in each country, and a product cannot be released to the market without regulatory approval. The development of a new drug usually takes 10 to 15 years, therefore clinical development of COVID-19 mRNA vaccines in one year is impossible without the strategic initiative of the government. In fact, in the United States, a national project called “operation warp speed” was organized to provide various supports for the rapid development of the COVID-19 vaccine. On the other hand, in Japan, the government has a history of not being proactive in vaccine development due to vaccine-related drug accidents. As such, the impact of government regulations has various effects on the clinical development of vaccines.

The second is the technological aspect. No matter how much regulatory and administrative support is provided, clinical development will not be successful unless the technology has reached the specifications for commercial application as a pharmaceutical technology. The profile of a new drug is assessed in non-clinical studies in animals and clinical studies in small groups to confirm whether it can demonstrate sufficient efficacy and safety in humans. In addition, APIs used in clinical trials must be manufactured in accordance with Good Manufacturing Practice (GMP), and in some cases, scale-up may cause the specifications to deviate from the standards, forcing a review of the manufacturing process. In particular, the mRNA vaccine is a new drug modality that had never been applied to a marketed drug before, and it can be inferred that more problem-solving would have been required than with ordinary modalities. Therefore, it is essential to discuss the technological aspects of BioNTech’s and Moderna’s mRNA vaccine research and development in order to examine the factors contributing to the rapid development success of both companies. This paper provides a detailed review of the technological and clinical development efforts undertaken by both companies through a comprehensive survey of published papers, patent applications, and information on clinical trials conducted.

The third is the investment aspect. BioNTech and Moderna are bio-ventures founded in 2008 and 2010, respectively. For a bio-venture to be successful in technology development, a lot of investment and the ecosystem to support it are necessary. A long-term investment is required especially in pharmaceutical R&D because new drug development requires a large amount of money and time. In this paper, I examine how BioNTech and Moderna were founded and how they obtained funding, and discuss the implications of the investment in vaccine development for both companies.

## 2. Administrative Aspect

To accelerate the development and distribution of COVID-19 vaccines and diagnostics, the US government initiated Operation Warp Speed (OWS) as a national project in May 2020 [9]. The project is funded to the tune of USD 10 million and conducted through a partnership between the Departments of Health and Human Services (HHS) and the Department of Defense (DOD) to enable rapid clinical development, approval, and mass production of COVID-19 vaccine [10,11]. The development of a new drug is usually conducted by manufacturing a small-scale drug substance and testing its efficacy and safety in a small number of patients in clinical trials. Clinical trials are conducted in phase 1, 2, and 3 and are managed using the stage-gate method [12]. This is an important methodology for pharmaceutical companies to mitigate risk in new drug development, which involves large clinical development costs and high attrition rates [13]. On the other hand, in the OWS, pharmaceutical companies were allowed to leverage data from other vaccines using similar technology platforms and to conduct non-clinical trials, usually conducted prior to clinical trials, in parallel with clinical trials [10,11]. In addition, unprecedented measures were taken to facilitate the procurement of equipment and materials needed for vaccine production, and the government guaranteed the purchase of large quantities of drugs for use in actual clinical settings before clinical trials were completed, which contributed significantly to accelerating the timeline for vaccine development [10,11].

In Japan, in order to promote the practical use of the COVID-19 vaccine, the Ministry of Health, Labour and Welfare (MHLW) has allocated a supplementary budget of approximately USD 1 billion and launched an urgent project for the development of a vaccine production system in 2020 [14]. This project supports the early establishment of a large-scale production system for the COVID-19 vaccine, and six companies were selected for the project. Subsequently, a supplemental budget of approximately USD 1 billion was additionally allocated to subsidize the cost of conducting vaccine clinical trials for the selected companies [14]. The Japan Agency for Medical Research and Development (AMED) has contributed approximately USD 1.5 billion to support the development of vaccines, therapeutic methods, basic technologies, equipment and systems against COVID-19 [15]. The number of supported projects is as many as 400, and the programs of six companies developing COVID-19 vaccines in Japan are also receiving support [15]. On the other hand, there is a historical background in Japan where the government has not been proactive in vaccine development, and there is an opinion that this has affected the delay in the development of the COVID-19 vaccine in Japan. Japan’s vaccine technology was high until the 1980s, but in the 1990s, the government was held liable for adverse reaction lawsuits caused by vaccines, and other drug-related lawsuits also affected the number of vaccinators, which significantly decreased [16]. Since then, a vaccine gap became the norm in Japan. Vaccines developed in Europe and the U.S. were approved several years to more than a decade later in Japan, and the motivation of pharmaceutical companies to develop vaccines declined [17]. The government was also reluctant to introduce new vaccines, for example, not approving vaccines tested in clinical by Japanese companies even though they had already been approved in the United States [17]. In the U.S., on the other hand, vaccines are viewed as a strategic military substance, and the U.S. military has supported vaccine research and development by biotech companies with subsidies from peacetime to secure vaccines [18]. These differences in administrative efforts are said to be reflected in the differences in the rapid development of the COVID-19 vaccine. However, as mentioned above, the Japanese government is also providing various supports for the rapid development of COVID-19 vaccines, therefore it is difficult to discuss differences in the speed of vaccine development solely on the basis of administrative issues.

## 3. Technological Aspect

The first successful introduction of mRNA into cells to make proteins was in 1971 [19]. In 1989, Dr. Malone efficiently introduced mRNA into cells with cationic lipids and produced proteins [20]. Inspired by the results of this experiment, Dr. Malone came up with the idea that proteins were made from mRNA in the body and used as drugs. He recorded this idea and this is said to have been the first idea of mRNA therapeutics [5]. However, the first mRNA vaccine was not approved as a drug until 2020, 30 years after it was conceived [21]. The reason why mRNA vaccines have taken so long to become commercially available is that there are a number of technical issues that need to be resolved. First is immune-stimulatory activity. A single-stranded RNA activates innate immunity through Toll-like receptors 7 and 8 [22,23]. This effect was greatly reduced by replacing uridine, one of the nucleic acids constituting the RNA, with pseudouridine [24]. The second is the optimization of RNA structure. 5’ cap structure is essential for RNA translation [25] and its artificial addition is required for mRNA vaccines. Poly(A) tail contributes to mRNA stability and translation efficiency [26] and optimization is necessary for mRNA vaccines to work. 3’ and 5’ untranslated regions (UTRs) are also important for mRNA stability and translation efficiency [27], therefore, must be optimized. The codon composition of the open reading frame affects mRNA translation efficiency, and GC-rich sequences are known to increase translation efficiency 100-fold compared to less GC-rich sequences [28]. Modification of these mRNA structures is necessary to improve mRNA vaccines. The third is the issue of delivery. mRNA is degraded by endonuclease and exonuclease in the blood, and its half-life is 7 h [29]. Therefore, when mRNA vaccines are introduced into the body, some methods to protect mRNA are needed. Currently, the most common method is to wrap mRNA in lipid nanoparticles (LNPs), which are also used in clinically applied mRNA vaccines [30]. The structure of LNP has been studied with many optimizations required for the type of lipids used as components and the balance of the formulation [31]. Fourth is manufacturing. Even if mRNA vaccines can be produced on a laboratory scale, they must be manufactured in large batches for human administration. Substances to be administered to humans must be manufactured in accordance with GMP. Scalability and cost-effectiveness are major issues for mRNA vaccines [32]. Fifth is clinical translation. The mRNA vaccine was first administered to humans in 2009 aimed as a therapeutic vaccine for cancer [33]. Since then, the majority of mRNA vaccines have been tested in clinical trials in the areas of oncology and infectious diseases. The challenges in clinical trials to date have been efficacy, administration route, and toxicity [34]. In addition, cancer vaccines are primarily for therapeutic use while infectious disease vaccines are for prophylactic use, therefore they have different criteria and challenges. Against this backdrop, the first mRNA vaccine did not receive regulatory approval until the end of 2020, and many challenges still remain in the clinical translation of mRNA vaccines.

Thus, the practical application of mRNA vaccines has faced many challenges from the research to the development stage, and only when the technologies to overcome these challenges are in place can they be applied clinically. Next, I will review how BioNTech and Moderna conducted their research and development and brought the COVID-19 mRNA vaccine to market, focusing on information from papers, patents, and clinical trials. In contrast, the case of Daiichi, which was developing a COVID-19 mRNA vaccine at the same time but has not yet received approval, will also be presented.

### 3.1. BioNTech

Ugur Sahin, chief executive officer of BioNTech, and Ozlem Tureci, chief medical officer of BioNTech, started their research on mRNA in the 1990s. They conducted research at universities and published papers and patents. Later, they wanted to apply their research to clinical practice, and in 2001 they founded Ganymed, a company specializing in antibody therapeutics for cancer. Then, they founded BioNTech in 2008.

In order to track BioNTech’s R&D, I searched PubMed for articles published by BioNTech on mRNA therapeutics and categorized them according to their contents (Figure 1). The largest number of papers dealt with the basic technology of mRNA therapeutics and its application in cancer therapy. Although fewer in number, research papers on infectious diseases other than COVID-19 have been published since 2017, and research papers on rare diseases have also been published in 2019. Papers on the basic technology of mRNA therapeutics have been published consistently since 2014, and the actual research period is considered to be several years before the publication of the papers, indicating that BioNTech has been continuously working on the development of mRNA therapeutics technology since the company was founded.

Although papers whose research dates are before the founding of BioNTech were not found in this search because the author affiliations are listed under university names, Sahin and Tureci’s group has been publishing epoch-making research results on the basic technology of mRNA therapeutics since the late 2000s. As mentioned above, the poly(A) tail and 3’UTR regions of mRNA are important for enhancing mRNA translational efficiency and stability. In a 2006 paper, Sahin, Tureci and colleagues identified poly(A) tail and 3’UTR sequences that enhance mRNA stability and translational efficiency [35]. In 2008, they developed a new antigen sequence encoded by mRNA that changes the subcellular distribution of the antigen and significantly increases CD8 and CD4 responses [36]. 5’cap structure is also important for mRNA translational efficiency and stability, and in 2010, they developed a new cap analog that increases the stability and translational efficiency of mRNA vaccines [37]. After the founding of BioNTech, they have continued to improve the cap structure, and have published papers on cap structure optimization [38,39] and quality control [40].

Weakening the immune response of RNA itself has been a major challenge for the practical application of mRNA vaccines. Dr. Katalin Kariko has been working on mRNA since 1989 and started working on mRNA vaccines with Prof. Weissman in 1997 [41]. In 2005, Kariko and colleagues found that changing one of the mRNA nucleic acids, uridine, to pseudouridine in which the uracil is attached via a carbon-carbon instead of a nitrogen-carbon glycosidic bond could greatly reduce the innate immune-activating effect of mRNA [24]. In the COVID-19 mRNA vaccine, the BioNTech and Moderna vaccines, which were approved with high clinical prophylactic efficacy, used N1-methyl-pseudouridine, while the CureVac mRNA vaccine, which was less effective in clinical use, did not. This difference is said to have contributed to the difference in clinical results [42]. Thus, the utilization of pseudouridine contributed significantly to the success of the mRNA vaccine. After discovering the effects of pseudouridine, Kariko gave an invited lecture in Mainz in 2013, where she met with Sahin and recommended the use of pseudouridine-modified mRNA to him. This led Kariko to work for BioNTech [43] and BioNTech was successful in bringing in Kariko, who has long experience and strong expertise in modifying mRNA for therapeutic use.

Since mRNA is unstable in vivo in its naked state, it is necessary to encapsulate it in liposomes or other media for delivery to the body. BioNTech has also focused on the research and development of LNPs and conducted the first intravenous nanoparticle delivery of mRNA vaccines in humans in 2014 [44]. After that, they have been working on nanoparticles that deliver mRNA and have published their research results [45,46].

For the practical application of mRNA vaccines, it is also important to overcome the manufacturing challenge. In order to distribute the product on the market as an approved drug, it is necessary to establish a mass production method that is different from small-lot production on a lab scale, and a GMP-compliant production method must be established for clinical trials. Sahin recognized the importance of this. BioNTech was able to produce approximately 10,000 doses annually prior to COVID-19, then they refined the process further and could scale it up to more than a billion doses with minor adaptations [47]. BioNTech has also recently published research results on how to efficiently remove double-strand RNA during mRNA purification and how to improve the stability of mRNA storage [48,49].

BioNTech’s technological developments in mRNA structure optimization, delivery, and manufacturing can be seen in their patent applications. Patents filed by BioNTech by the end of the year 2020 were searched on Espacenet where the applicant is BioNTech and all text fields contain mRNA (Figure 2). Although BioNTech was founded in 2008, patents for mRNA-related technologies under the BioNTech name began to be filed earlier, in 2002. The number of patent applications has been consistently growing until 2020, indicating that the company has been actively developing and licensing technologies for about 20 years. Of these, patents that appeared to be related to the optimization of mRNA structure were filed in the years 2005, 2006, 2014, 2015, and 2018, and patents that appeared to be related to delivery technology including LNP were filed in 2009, 2012, 2013, 2014, 2015, 2017, 2018, and those that appeared to be related to formulation and storage technology were filed in 2016, 2018, 2019 and 2020. It can be inferred that the company has been establishing technologies related to mRNA modification and delivery since the 2000s, and more recently has been focusing on technologies closer to the product, such as formulation.

Clinical trials conducted by BioNTech were searched using the Clinicaltrials.gov website (https://clinicaltrials.gov/, accessed on 9 October 2022) for studies with BioNTech as the lead sponsor using advanced search and confirmed from the study details that the study was for mRNA therapeutics. The extracted studies were categorized by disease type (infectious diseases (other than COVID-19), tumor, and COVID-19 vaccine) (Figure 3). BioNTech initiated clinical trials in 2012. Until 2019, all trials were in oncology. From 2020, clinical trials for the COVID-19 vaccine began alongside oncology, with 2020 and 2021 representing the highest percentage of the number of clinical trials. This process suggests that BioNTech has applied its clinical knowledge and experience accumulated in oncology to infectious diseases. BioNTech recognizes the importance of conducting sufficient preclinical research to elucidate the mechanism, etc., and then translating the results into clinical settings, and their efforts are introduced in their review [50]. In addition, BioNTech has been working since 2015 to develop vaccines against infectious diseases in parallel with cancer [41]. Kariko has proven the efficacy of mRNA vaccines against the Zika virus in animals in 2017 [51]. These R&D experiences suggest that BioNTech was also well-experienced and knowledgeable in developing infectious disease vaccines at the time of COVID-19’s emergence. Sahin said in his interview that he knew the extraordinary potency of their mRNA vaccine platform; therefore, he decided to develop the COVID-19 vaccine and discussed with Tureci how to carry this out the next day he heard the news of the COVID-19 pandemic [47]. He mobilized the team in January 2020 and quickly started the COVID-19 vaccine project at BioNTech [47].

### 3.2. Moderna

As described, Kariko and colleagues found that modifying the uridine residue of mRNA can significantly suppress the inflammatory response caused by native mRNA while maintaining its ability to be translated into protein [24]. Moderna’s founder, Derrick Rossi read the paper in 2005 while working on stem cell research and recognized it as a breakthrough technology. In 2007–2009, Rossi, then an associate professor at Harvard University, found that modified mRNAs using the modifications found by Kariko and colleagues could be introduced into adult cells and transformed into embryonic stem cells [52]. He saw great potential in mRNA therapeutics and founded Moderna in 2010. At the time, Rossi had no idea that mRNA could be used as a vaccine.

In order to track Moderna’s R&D, I searched PubMed for articles published by Moderna on mRNA therapeutics and categorized them according to their contents (Figure 4). 

Moderna’s paper publications started in 2014, with more than 10 papers published per year starting in 2017. Papers on mRNA therapeutics platform technology were consistently published starting in 2014, with the highest number of papers published through 2019. Moderna’s total publication number was low up to 2016, while patent applications searched on Espacenet peaked in 2013 and then decreased considerably in 2016 (Figure 2). It is possible that the company refrained from publication until they completed the development of patentable technologies therefore the actual research may have been conducted much before the year when the paper was published, so the timing of the research should be carefully considered. In any case, the contents of the papers published so far indicate that much research has been conducted on mRNA structure optimization, mRNA delivery techniques, and manufacturing. Regarding mRNA structure, papers on the optimization of open reading frame [53] and 5’UTR sequence [54] were published. Many papers were published on delivery including review articles, and studies on the lipid composition of LNP [55,56] and size [57] were published. Regarding production, a method for mRNA purification [58], a synthesis method with less byproduct [59], and a study on temperature stability [60] were presented.

Clinical trials conducted by Moderna to date are published by Moderna on its web page (https://trials.modernatx.com/search-results/?PageIndex=0, accessed on 2 October 2022). Moderna initiated the first clinical trial in 2015 (Figure 5).

Looking at patent applications, applications began in 2010, when the company was founded, with the largest number of applications in 2013, followed by a decline in the number of applications, and once reached zero in 2016 (Figure 2). This suggests that Moderna had filed patents on mRNA vaccine technology for clinical trials roughly by 2015 and had moved from research to the clinical development phase (Figure 5). Until 2019, the largest number of clinical trials were for infectious diseases other than COVID-19. Clinical trials for tumors and rare diseases were also initiated. Clinical trials for the COVID-19 vaccine began in 2020 and continued until 2022. The COVID-19 vaccine accounted for the largest proportion of the number of clinical trials until 2022. It can be seen from their publication that Moderna already had experience with mRNA vaccines as infectious disease vaccines at the time of COVID-19’s emergence. In 2017, Moderna published the results of a clinical trial against the influenza virus [61]. As of 2021, mRNA vaccine trials against infections other than COVID-19 are being conducted with 18 compounds, 17 in phase 1 and 1 in phase 2, 13 of which are Moderna’s compounds [62]. These data show how experienced Moderna was in the clinical development of mRNA vaccines against infectious diseases other than COVID-19. It is strongly inferred that Moderna had already established its expertise in vaccine development at the time of pandemic emergence through the development of technologies related to mRNA structure, delivery, and manufacturing, as well as the clinical development of mRNA vaccines, especially vaccines against infectious diseases.

### 3.3. Daiichi

Daiichi has developed an mRNA vaccine against COVID-19 in Japan [63]. As of November 2022, clinical development has not been completed.

In order to track Daiichi’s R&D, I searched PubMed for articles published by Daiichi and Espacenet for patents applied by Daiichi on mRNA therapeutics. However, I could not find any publication or patent. The only publication identified on the web was a mini-review (not indexed by PubMed). According to the review, Daiichi uses a proprietary technology for the production of LNP used for mRNA vaccines [64]. However, since no patent applications were identified and no original papers were published, it is unlikely that research and development have been active for many years. Next, I searched for clinical development information. The clinical trials of Daiichi’s mRNA vaccine were extracted from the clinical trial information available on the website (https://www.daiichisankyo.co.jp/files/rd/pipeline/index/pdf/FY2022Q1_Pipeline_J.pdf, accessed on 2 October 2022), and the content and start dates of the clinical trials were identified from the press releases. Daiichi initiated two clinical trials of mRNA therapeutics, one in 2021 and one in 2022, both of which were clinical trials of the COVID-19 vaccine. No clinical trials with mRNA therapeutics had been conducted prior to these trials. Unlike BioNTech and Moderna, it was not believed that Daiichi had any experience and accumulation in mRNA vaccine research and development prior to COVID-19 emergence.

## 4. Investment Aspect

Neither BioNTech nor Moderna had any products on the market until the approval of the COVID-19 mRNA vaccine. Both companies were able to spend a large amount of R&D costs for about 10 years since their establishment because they acquired large investments and payments to cover the money spent. In the case of BioNTech, the company received a large investment from the Struengmann brothers, famous investors in Germany, at the time of its founding. In 2008, Thomas Struengmann met Sahin and Toreci, liked their enthusiasm and technology, and decided to invest in them [65]. BioNTech obtained USD 270 million of series A financing in 2018 [66]. According to their annual report, the loss before tax was about 180 million euros in 2019 [67], suggesting that the expectation of the technology attracted large investments and supported funding while there was a large cost of research and development. It should be also noted that Sahin and Tureci are serial entrepreneurs who founded Ganymed in 2001 for the development of cancer antibody drugs. Ganymed was acquired by Astellas Pharma in 2016 for 420 million euros [68].

In the case of Moderna, when Derrick Rossi was convinced of the potential of mRNA therapeutics, he went to Robert Langer of MIT, a well-known chemical engineer and entrepreneur, and Noubar Afeyan, a venture capitalist, and asked for their support. They believed that mRNA therapeutics could be used not only for regenerative medicine but also for the treatment of many diseases, and they joined Moderna’s founding team. Soon after its launch, Moderna’s technology attracted much attention, resulting in large contracts and investments from major pharmaceutical companies. In 2013–2014, Moderna raised USD 1.9 billion from AstraZeneca, Alexion and Merck including payment and investment [69]. In 2013, Moderna raised USD 450 million in a financing round, shattering all previous records for a privately held biotech company [70]. It is clear that the promise of mRNA therapeutics and the corresponding bountiful investment supported Moderna’s R&D funding.

## 5. Conclusions

Clinical development and approval of pharmaceutical products are strongly influenced by the regulatory affairs of each country. Therefore, it is important to examine the administrative influence on the rapid development of COVID-19 mRNA vaccines in order to consider the factors that contributed to their success. The reasons for the success of rapid development of Moderna and BioNTech’s mRNA vaccines are often discussed in terms of the accelerated regulatory process and the history of vaccine development. In the United States, the OWS, a public–private partnership program, was promoted under strong national leadership, and this was a major boost to the rapid development of the COVID-19 mRNA vaccines. In Japan, although the government had been reluctant to support vaccine development in the past due to a history of drug-related side effects, the government spent a large amount of money to support the rapid development of vaccines after the COVID-19 outbreak. In addition, some firms succeeded in rapid vaccine development while others did not, even under the same national regulations. These suggest that administrative influences contribute to a certain degree to the success of rapid development, but this alone may not be a sufficient explanation.

What I learned from the comparison of the companies that succeeded in the rapid development of mRNA vaccines or did not is that there was an overwhelming difference in technological readiness. A comprehensive review of publications, patents, and clinical trials revealed that BioNTech and Moderna had conducted extensive research and development on mRNA therapeutics prior to the advent of COVID-19. Even before BioNTech was founded, in the late 2000s, Sahin and Tureci had developed a series of technologies that were important for the practical application of mRNA vaccines. The key technologies for optimizing the poly(A) tail and 3’UTR were published in 2006, the discovery of a sequence for enhancing antigen presentation was published in 2008, and the optimization of the cap structure was published in 2010. After BioNTech was established, papers on the basic technology of mRNA therapeutics have been consistently published since 2014, indicating that the company has been working on mRNA structure optimization research for many years. BioNTech has been also working to establish the LNP technology necessary for mRNA delivery. In 2014, they administered mRNA using LNP in a clinical setting for the first time. Establishing a GMP-compliant mass production method is necessary for clinical use, and Sahin recognized the importance of this and had already established a system capable of producing 10,000 doses per year prior to COVID-19. The accumulation of long-term technological development is evident in BioNTech’s patent applications. Already in 2002, patent applications under BioNTech’s name began to be filed, and the number of applications was consistently increasing until 2020. Additionally, they invited Kariko, one of the pioneers in the development of mRNA vaccine technology, to further strengthen the capability. BioNTech has accumulated clinical development experience mainly in the area of oncology vaccines. They began clinical trials of mRNA vaccines in 2012, and by 2019, before the emergence of COVID-19, they had already conducted 13 clinical trials. In parallel, BioNTech has been working on infectious disease vaccines since 2015, proving the efficacy of mRNA vaccines against the Zika virus at the animal level in 2017. Based on these accumulated experiences, at the time of the emergence of COVID-19, BioNTech was ready to develop and commercialize an mRNA vaccine against COVID-19, and they made an immediate decision to work on its development. Tureci said in her interview, “More than two decades of technology development and sound scientific research prepared us for developing the COVID-19 vaccine and enabled us to do it in such a short time” [47]. It is apparent that years of research and development of mRNA vaccines, particularly cancer vaccines, existed as a predisposition for the rapid development of the COVID-19 mRNA vaccine at BioNTech.

Moderna also possessed much experience in technology development and clinical trials of mRNA vaccines. Their publication revealed that they had been performing research on the sequence optimization of ORF and 5′UTR regarding mRNA structure, LNP optimization regarding delivery, and mRNA manufacturing. Many patents were filed from 2010–2015, prior to the time of publication, and the number of applications peaked in 2013 and declined thereafter, once dropping to zero in 2016. Presumably, Moderna developed the technologies for the clinical application of mRNA vaccines for this time frame and prepared for clinical trials, which began in 2015. Notably, Moderna had conducted many clinical trials in infectious diseases prior to the advent of COVID-19. Among mRNA vaccine trials against viruses other than COVID-19 by 2021, many of them were conducted by Moderna. Moderna published clinical trial results of the mRNA vaccine against the influenza virus in 2017. It is speculated that Moderna had learned a lot of things from their technology development and clinical trials before they developed the COVID-19 vaccine although the first approved mRNA vaccine was for COVID-19.

In contrast, I could not identify any accumulation of technology development and clinical trials of mRNA vaccine before the advent of COVID-19 in Daiichi. Although the publication confirmed that an original technology is used for the lipid of LNP, no international patents were filed, and no other publications or patent applications for mRNA vaccines were identified. The clinical trial of the mRNA vaccine was also the first for COVID-19, and the experience of technology and clinical development is significantly less than that of BioNTech and Moderna. It is easy to infer that this overwhelming difference in technology readiness has affected the difference in the speed of development of the COVID-19 vaccine.

Having a lot of experience in technology development and clinical trials is especially important in drug development. This is because new drug development using human subjects is conducted under strict regulations, and if a problem arises in clinical trials, such as efficacy, safety, or formulation, the drug candidate compound must be reselected or the manufacturing process changed in the research stage, causing a significant delay of the development period. Without a doubt, BioNTech and Moderna were familiar with mRNA vaccine technology by developing it themselves, and with experience in many clinical trials in viruses and disease areas other than COVID-19, they knew what the problems were and how to overcome them when administered to humans. The impact of investment is important in supporting this long period of R&D. Pharmaceutical R&D typically takes 10–15 years, and new modalities require more time for initial technology development. Neither BioNTech nor Moderna had a product on the market before COVID-19 vaccine approval. During that time, large R&D expenses kept the income statement in the red, with BioNTech’s loss in 2019 amounting to about 180 million euros. However, BioNTech had received substantial investment from business angels since its establishment and had also raised a wealth of venture capital funding. Moderna had the support of a prominent researcher and a venture capitalist with extensive entrepreneurial experience from its establishment. Moderna had also received much joint research funding and investment from several pharmaceutical companies that were interested in their technology and had successfully raised a large amount of money from the market as well. When Rossi founded Moderna, he did not imagine that mRNA drug technology could be used for vaccines against infectious diseases, and BioNTech has been refining its mRNA vaccine technology mainly in the oncology area. As such, innovative technologies often do not have their optimal applications in the early stage. Nevertheless, when there is interest in the innovativeness of a technology, angels and venture capitalists invest heavily from the early stage, creating an ample financial base for deep tech to grow. In the U.S. in particular, there is an abundance of risk money to invest in deep tech, and a culture of investing in and nurturing promising technologies from the early stages and the ecosystem that supports it are well established. In Japan, on the other hand, the amount of risk money used for venture investment is a hundredth of that in the U.S. [71], and the venture ecosystem has not taken root. Although the number of start-ups temporarily increased in the 2000s due to the 1000 University Ventures Plan, most of them are very small companies with a few employees, and those with large market capitalization or commercial success are rare [72]. Therefore, even new technologies such as mRNA therapeutics are currently being developed by existing pharmaceutical companies such as Daiichi using their own funds. This difference in the start-up ecosystem is thought to be the reason for the difference in the incubation of deep tech that solves sudden social issues and the difference in the response in case of emergency.

The COVID-19 pandemic was a symbolic event in the era of VUCA. The development of a variety of advanced technologies is essential to tackle public health challenges. However, the practical application of such advanced technologies takes a long time and requires a large investment. By comprehensively reviewing the R&D history of BioNTech and Moderna and the policy and investment perspectives surrounding it, this paper describes the importance of technology incubation in the long term and the need for a venture ecosystem to support it.

## Figures and Tables

**Figure 1 pathogens-11-01469-f001:**
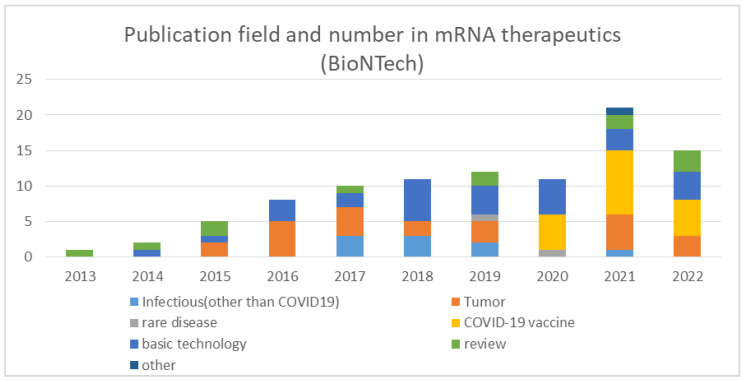
The number and field of paper publication in mRNA therapeutics by year by BioNTech.

**Figure 2 pathogens-11-01469-f002:**
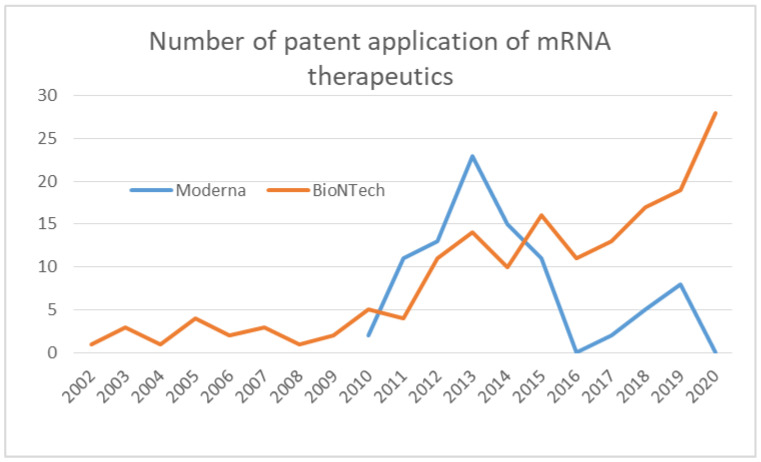
The number of patent application in mRNA therapeutics by year by BioNTech and Moderna.

**Figure 3 pathogens-11-01469-f003:**
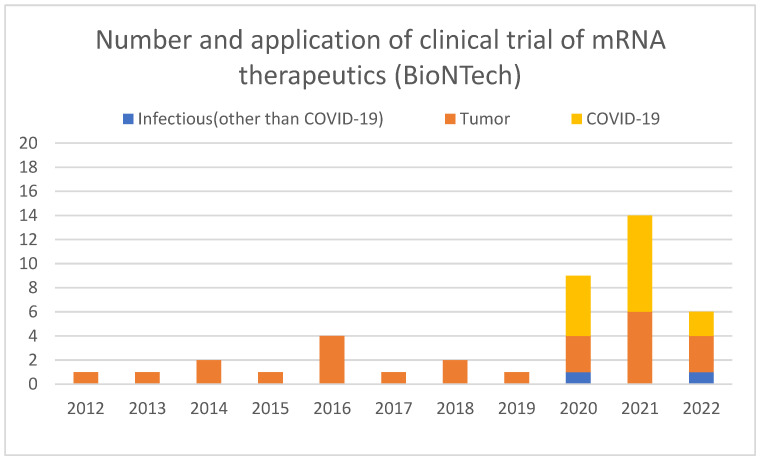
The number and application of clinical trial of mRNA therapeutics by year by BioNTech.

**Figure 4 pathogens-11-01469-f004:**
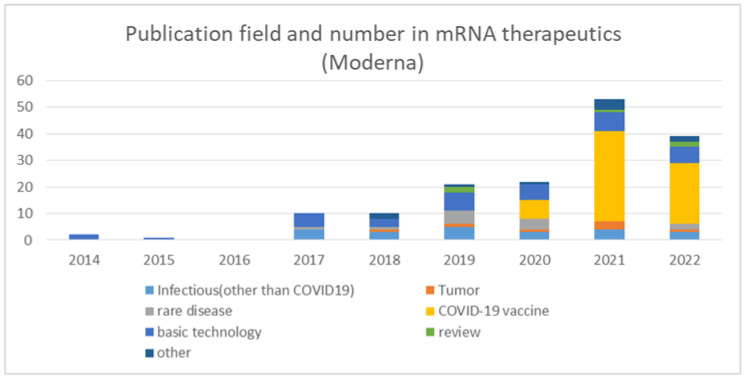
The number and field of paper publication in mRNA therapeutics by year by Moderna.

**Figure 5 pathogens-11-01469-f005:**
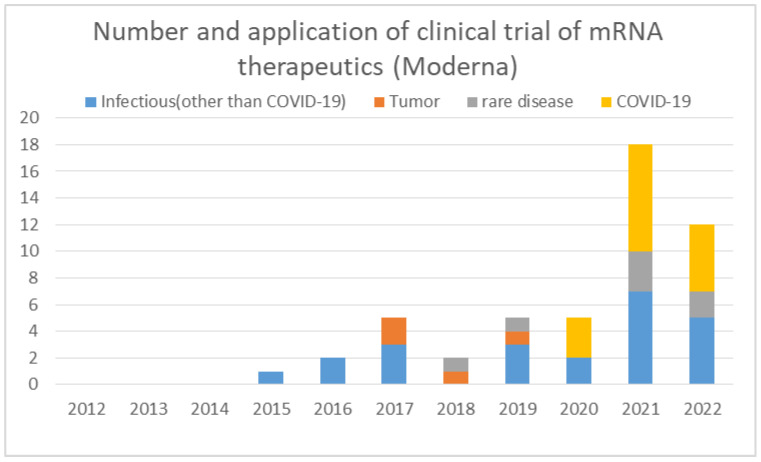
The number and application of clinical trial of mRNA therapeutics by year by Moderna.

## Data Availability

The data supporting results of Figure 1 and Figure 4 were obtained by the search of PubMed (https://pubmed.ncbi.nlm.nih.gov/). The data supporting results of Figure 2 were obtained by the search of Espacenet (https://worldwide.espacenet.com/). The data supporting Figure 3 were obtained by the search of ClinicalTrials.gov (https://clinicaltrials.gov/). The data supporting Figure 5 were obtained from Moderna’s homepage (https://trials.modernatx.com/search-results/?PageIndex=0).

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
