# Peer review of "Nurturing Deep Tech to Solve Social Problems: Learning from COVID-19 mRNA Vaccine Development"

_pathogens, 2022, doi:10.3390/pathogens11121469_

Round 1

Reviewer 1 Report (Previous Reviewer 1)

A revised version of the manuscript was substantially improved. Nonetheless, it still seems to be a little bit rush out. Many typos are present in the text, e.g. line 89 (COVD-19), line 250 (BiNTech), etc. In my first review report, I wanted to explain the VUCA term, but not only the abbreviation, I meant rather the "concept" of VUCA. Overall I think that the author should carefully proofread the manuscript in detail once again before it can be accepted for publication. 

Author Response

To Reviewer 1,

Thank you very much for reviewing my manuscript. In accordance with your comments, I added the concept of VUCA to the beginning of Introduction (see attached file). I carefully proofread the manuscript and corrected twenty typos and grammatical errors (see attached file in detail). I would greatly appreciate your confirmation.

Reviewer 2 Report (New Reviewer)

The review manuscript-2081026 was well written, organized and highly readable, the description of the success of mRNA vaccine development against COVID-19 by BioNTech and Moderna is accurate, and clear, the factors in administrative support, mRNA technology development and venture investment are inspiring, the lessons learnt here should benefit better preparation for next public health issues, and I really enjoy reading this review article. I have no significant comments on this work.

I have only two minor comments.

 Figure 3: it is recommended to remove legend for rare disease since there is no any clinical trial corresponding to this category.

Figure 6: it is recommended to delete figure 6, since only two clinical trials were initiated and they were already described at lines 370 and 371.

Author Response

Dear Reviewer 2,

Thank you very much for reviewing my manuscript. In accordance with your comments, I removed the legend for rare disease from Figure 3 and deleted Figure 6 (see attached file). I would greatly appreciate your confirmation.

This manuscript is a resubmission of an earlier submission. The following is a list of the peer review reports and author responses from that submission.

Round 1

Reviewer 1 Report

In this research article entitled "Nurturing deep tech to solve social problems: Learning from COVID-19 mRNA vaccine development", Ryo Okuyama carried out systematic literature searches to compare the founding history of three biotechnological companies: BioNTech, Moderna, and Daiichi (Japan). The paper contains some interesting facts which were generally little known and is relatively well-written and logically structured, nonetheless, in my humble opinion, this is not a research article but rather a review, as no primary data were produced/analyzed, only bar plots of existing literature, patent applications, and clinical studies counts were made. Therefore I must reject the article in its current form. My suggestions are as follows:

1.) Change the article type to Review and substantially expand the overall number of cited References (should be at least 50 + try to avoid blog post sources, e.g. "Newsweek Japan; WHY JAPAN LOST THE VACCINE RACE. 17, Nov. 2020 https://www.newsweekjapan.jp/sto-356 ries/world/2020/11/post-95015_1.php", mainly primary PUBMED research papers should be cited)

2.) if you want to publish an "Article", your paper must contain at least some kind of statistical analysis of obtained results, i.e. are there significant differences between Moderna, Pfizer, and Daiichi histories?

Please don't take it as hard criticism of your work, the text is interesting and readable, and I appreciate your effort in categorizing existing literature on this theme. Personally, I think it could be a nice review for this journal (Pathogens) once several (20-30) references will be added and the Results and Discussion section will be expanded. Unfortunately, I must recommend the rejection of the manuscript in its current state. 

Further points/suggestions for improvement:

 - VUCA term should be explained

- Line 119 ... I searched; Line 127 ... We searched

- In the sentence "Rather, it is believed that the engineers and investors who realized that the technology had many possibilities, although the application of the technology was not clear, invested in and incubated the technology from its early days, which increased the maturity of the technology and ultimately enabled the rapid application of the technology to COVID-19 vaccines." ... word technology is repeated 5x